# High Prevalence of *Staphylococcus aureus* Enterotoxin Gene Cluster Superantigens in Cystic Fibrosis Clinical Isolates

**DOI:** 10.3390/genes10121036

**Published:** 2019-12-12

**Authors:** Anthony J. Fischer, Samuel H. Kilgore, Sachinkumar B. Singh, Patrick D. Allen, Alexis R. Hansen, Dominique H. Limoli, Patrick M. Schlievert

**Affiliations:** 1Stead Family Department of Pediatrics, University of Iowa Carver College of Medicine, Iowa City, IA 52242, USA; sachinkumar-singh@uiowa.edu (S.B.S.); Patrick.D.Allen@dmu.edu (P.D.A.); alexis-hansen@uiowa.edu (A.R.H.); 2Department of Microbiology and Immunology, University of Iowa Carver College of Medicine, Iowa City, IA 52242, USA; samuel-kilgore@uiowa.edu (S.H.K.); dominique-limoli@uiowa.edu (D.H.L.); patrick-schlievert@uiowa.edu (P.M.S.)

**Keywords:** cystic fibrosis, *Staphylococcus aureus*, superantigen, enterotoxin gene cluster, MRSA

## Abstract

Background: *Staphylococcus aureus* is a highly prevalent respiratory pathogen in cystic fibrosis (CF). It is unclear how this organism establishes chronic infections in CF airways. We hypothesized that *S. aureus* isolates from patients with CF would share common virulence properties that enable chronic infection. Methods: 77 *S. aureus* isolates were obtained from 45 de-identified patients with CF at the University of Iowa. We assessed isolates phenotypically and used genotyping assays to determine the presence or absence of 18 superantigens (SAgs). Results: We observed phenotypic diversity among *S. aureus* isolates from patients with CF. Genotypic analysis for SAgs revealed 79.8% of CF clinical isolates carried all six members of the enterotoxin gene cluster (EGC). MRSA and MSSA isolates had similar prevalence of SAgs. We additionally observed that EGC SAgs were prevalent in *S. aureus* isolated from two geographically distinct CF centers. Conclusions: *S. aureus* SAgs belonging to the EGC are highly prevalent in CF clinical isolates. The greater prevalence in these SAgs in CF airway specimens compared to skin isolates suggests that these toxins confer selective advantage in the CF airway.

## 1. Introduction

Cystic fibrosis (CF) is a common lethal genetic disease, which results in chronic airway infections, irreversible bronchiectasis, and respiratory failure. *Staphylococcus aureus* is the most prevalent bacterial pathogen in children with CF [1], and is present in ≈70% of all individuals with CF in the United States. Although *Pseudomonas aeruginosa* is the predominant pathogen in older patients, *S. aureus* is the most common bacterial species in patients with CF under age 24 [2]. Unlike *P. aeruginosa*, which has multiple effective treatments to eradicate early infections and control chronic infections [3,4,5], *S. aureus* infections can be difficult to control with antibiotics [6]. CFTR modulator drugs may help prevent incident infections with *S. aureus*, but they are unlikely to eliminate chronic *S. aureus* infections [7,8]. Understanding how *S. aureus* infects and persists in the CF airway is critically important, as these infections may increase the risk of subsequent disease progression [9,10].

We hypothesized that *S. aureus* isolates in the CF airway would share common virulence properties. Some readily visible phenotypes such as hemolysis, pigmentation, and protease secretion could enable *S. aureus* to elude host defenses. People with CF are commonly treated with antibiotics; resistance to antibiotics may be occur under the selective pressure of antibiotic exposure. Another potential mechanism enabling *S. aureus* to establish chronic infection is the secretion of toxins that misdirect the immune response. *S. aureus* produces a large number of secreted toxins that may be critical for establishing infections [11]. These include 18 unique superantigens (SAgs), secreted toxins that bind both the T cell receptor and major histocompatibility complex molecules on antigen presenting cells [12,13].

Some SAgs are well known for their roles in acute infection. In the extreme example of toxic shock syndrome, the SAg TSST-1 cross-links T cells and antigen presenting cells, stimulating massive cytokine release and blocking the immune system from developing lasting immunity [12,13]. By contrast, the enterotoxin gene cluster (EGC), an element encoding six staphylococcal enterotoxin (SE) and SE-like SAgs G, I, M, N, O, and U, is generally associated with long term mucosal colonization. The EGC is present in between 50% to 70% of isolates from individuals with nasal carriage of *S. aureus* [14,15]. These EGC toxins can stimulate T cell proliferation [16], yet neutralizing antibody response to these toxins is surprisingly poor [17]. While these toxins have been associated with asymptomatic colonization, experimental studies in rabbits show that EGC SAgs may play crucial for infections such as endocarditis [18]. 

It is not clear what role SAgs play in CF respiratory infections. EGC SAgs are prevalent in clinical isolates of *S. aureus*; a recent European study showed that 57% of CF isolates harbored at least one gene belonging to the EGC [19]. This is similar to the prevalence of EGC in isolates from a cohort of patients with atopic dermatitis in the United States and Europe [20,21]. Our study had two goals: To determine whether these SAgs are as prevalent in CF isolates in the United States, and to determine if the superantigens were associated with methicillin resistance, which is common in the United States and has been linked to worse outcomes [10,22].

## 2. Materials and Methods 

Ethics Statements: All bacterial isolates examined in this study were de-identified when they were supplied to the research team. The University of Iowa Institutional Review Board (IRB) approved specimen collection after obtaining informed consent under approval numbers 200311016 and 200803708.

Sources of Clinical Isolates: *University of Iowa Cystic Fibrosis Biobank.* 77 de-identified *S. aureus* clinical isolates were obtained from the University of Iowa Hospitals and Clinics clinical laboratory following CF clinic visits made between 12 December 2011 and 20 July 2012. Specimens were obtained from both adult and pediatric patients. 

*CF Biospecimen Registry (CFBR) at Emory and Children’s Center for Cystic Fibrosis*: 20 *S. aureus* isolates from people with CF were collected between 1 January 2012 and 31 December 2013. These human subject samples were provided by the CF Biospecimen Registry at the Children’s Healthcare of Atlanta and Emory University CF Discovery Core courtesy of Arlene Stecenko. The Emory University IRB has approved collecting and banking of these specimens after obtaining informed consent. 

*The Geisel School of Medicine at Dartmouth University*: The Hogan laboratory at the Geisel School of Medicine at Dartmouth University graciously provided 12 deidentified *S. aureus* isolates from the Dartmouth CF Translational Research Core. These isolates were obtained from adult patients with CF between 1 January 2015 and 31 December 2017 with support by the CF Foundation RDP grant STANTO15R0. 

Bacterial Phenotypes: Clinical isolates of *S. aureus* were streaked onto tryptic soy agar (TSA) and blood agar to examine colony size, color, and hemolysis pattern. We streaked colonies onto milk agar to score for secreted protease. Beta-toxin was scored by partial lysis on sheep blood agar; alpha-toxin by complete lysis on rabbit blood agar. Oxacillin resistance was determined by growth on Mueller–Hinton agar with 4% NaCl in the presence or absence of 6 µg/mL oxacillin at 33–35 °C. We examined for chloramphenicol, tetracycline, or erythromycin resistance by presence or absence of growth with 10 µg/mL of each antibiotic.

Superantigen Testing: Clinical isolates of *S. aureus* were tested for the presence or absence of SAg genes using PCR of genomic DNA preparations following a published protocol with appropriate positive and negative controls for each of the SAgs [23]. PCR primers are listed in Appendix A. 

Statistical Analysis: We determined the prevalence of SAgs as the number of subjects positive for a given SAg divided by the total number of subjects analyzed. In subgroup analysis, we compared SAg prevalence in subjects with a single culture vs. those with multiple cultures using Fisher’s exact test. We compared the proportions of MRSA and MSSA isolates that were positive for each of the SAgs using Fisher’s exact test. To measure the strength of the association, we calculated an odds ratio to determine the increase in odds that the individual SAg would be present in MRSA compared to MSSA. Odds ratios were calculated using conditional maximum likelihood estimate with the fisher.test command in R. *P* < 0.05 was considered statistically significant. We did not adjust for multiple comparisons. To determine whether MRSA and MSSA have distinct complements of SAgs, we performed unsupervised hierarchical clustering of the University of Iowa Biobank isolates based on the presence of SAg genes. We used R Studio version 0.98.1085 or SAS version 9.4 for statistical testing.

## 3. Results

### 3.1. S. aureus Specimens From Patients With CF Are Heterogeneous in Phenotypic Appearance

We obtained 77 clinical isolates of *S. aureus* from adult and pediatric patients with CF at the University of Iowa. These specimens were obtained from *N* = 45 patients in visits between 12 December 2011 and 20 July 2012. The median age of these patients was 15.75 years as of the date of their last culture (IQR 8.34–26.89, range 5.27–58.66). Between 1 and 7 specimens were obtained per subject (Appendix A), with 28 subjects having a single culture. 44 of these specimens were from sputum samples and 33 were from oropharyngeal swabs. 

#### 3.1.1. Colony Morphology

In chronic airway infections, CF pathogens like *Pseudomonas aeruginosa* diversify through genetic mutations [24]. However, if phenotypes are required for survival in the CF airway, these features may be found with increased frequency. Therefore, we examined the *S. aureus* isolates from patients with CF for colony phenotypes (Table 1). *S. aureus* normally expresses staphyloxanthin, a golden pigment that protects against host-derived oxidants [25]. We found that many isolates were hypopigmented: 13 were white, 37 were yellow, and 27 were gold. 

Protease secretion is considered a virulence factor in skin infections [26], and protease production could be damaging to airways. Therefore, we tested for secreted protease by examining milk agar plates for zones of clearance. 23 of the isolates had distinct zones of clearance consistent with protease secretion, 13 had small or faint zones of clearance, and 41 isolates exhibited no clearance of milk agar in this assay. There was a strong correlation between hypopigmentation and protease activity. Protease activity, as determined by clearance of milk agar, was detected in 85% of white colonies but 7.5% of gold colonies (*P* < 0.001). 

Another characteristic of *S. aureus* is hemolysis, a phenotype that is linked to alpha and beta hemolysin toxins. Previous studies of bacteria deficient in alpha toxin reveal its importance in cellular and animal models of CF [27,28]. We tested for the activity of these toxins by hemolysis patterns on sheep and rabbit blood agar plates. Alpha toxin (encoded by *hla*) activity was observed in the majority of specimens, whereas beta-toxin (*hlb*) activity was observed in 40% of specimens.

#### 3.1.2. Antibiotic Resistance

CF pathogens are under selective pressure from antibiotic treatment. We determined MRSA status of these isolates by growth on Mueller–Hinton agar in the presence of 6 µg/mL of oxacillin. 28 isolates derived from 19 individuals were phenotypically resistant to oxacillin. 49 isolates from 30 individuals were methicillin-susceptible *S. aureus* (MSSA). Four subjects had isolates of both MRSA and MSSA. Because macrolides and tetracyclines are commonly prescribed to patients with CF [2], we hypothesized that the *S. aureus* isolates would be resistant to these antibiotic classes, but remain susceptible to antibiotics that are not routinely given. Each isolate was grown on TSA containing either erythromycin, tetracycline, or chloramphenicol. Chronic azithromycin is routinely prescribed at the University of Iowa CF center [8]. The vast majority of isolates from the University of Iowa (75/77) exhibited erythromycin resistance, but tetracycline resistance was less common (12/77). Within the University of Iowa collection, the two isolates susceptible to erythromycin were obtained as oropharyngeal cultures.

### 3.2. High Prevalence of Enterotoxin Gene Cluster Genes in S. aureus Isolated From Patients With Cystic Fibrosis

*S. aureus* encodes a variety of secreted toxins, including bacterial superantigens (SAgs). We hypothesized that there would be similar heterogeneity in *S. aureus* secreted toxins. Using previously described methods [23], we assessed for the presence of 18 unique SAgs. Among these toxins, the most common were genes belonging to the enterotoxin gene cluster (EGC), including *seg*, *sel-i*, *sel-m*, *sel-n*, *sel-o*, and *sel-u*. These genes were highly prevalent in isolates of *S. aureus* from patients with CF. All six EGC genes were identified in 37 of the 45 patients examined. 97.8% of patients within this cohort grew *S. aureus* that encoded at least one member of the EGC (Table 2). The genes encoding EGC toxins were significantly more prevalent in CF specimens compared to the classic *S. aureus* SAg toxic shock syndrome toxin-1(TSST-1; gene *tstH*), which was present in 11.7% of isolates.

Because some subjects within the University of Iowa cohort had multiple cultures, there may be greater opportunities to identify specific bacterial genes within these subjects. Therefore, we compared the prevalence of each toxin in subjects with multiple cultures vs. those with one culture. We identified *sea* and more frequently in patients with repeated cultures. Genes encoding EGC toxins were highly prevalent in both groups. We observed no statistically significant differences in age or culture source between groups.

#### 3.2.1. EGC Prevalence in MRSA and MSSA

We hypothesized that the genes encoding secreted toxins may be associated with either methicillin susceptibility or resistance. To determine whether MRSA isolates had specific toxin signature(s), we performed hierarchical clustering based on the presence or absence of toxins (Figure 1). MRSA isolates were distributed widely in this analysis and often shared the same toxin profile as MSSA isolates. We separately tested whether individual toxins were associated with MRSA or MSSA (Table 3). None of the MRSA isolates were positive for *tstH*, consistent with previous observations that TSST-1 is generally associated with MSSA [29]. MSSA was more likely than MRSA to be positive for *sel-p* (*P* = 0.03). While *sel-p* and *tstH* were more common in MSSA, *sel-x* was more common in MRSA. However, no combination of SAgs was perfectly predictive of methicillin resistance, and genes encoding EGC toxins were prevalent in both MSSA and MRSA isolates. 

#### 3.2.2. EGC Prevalence in *S. aureus* From Other U.S. CF Centers

We considered the possibility that the high prevalence of *S. aureus* encoding EGC was due to geographic sampling. To address this possibility, we obtained 12 *S. aureus* isolates from adults with CF at Dartmouth University and 20 *S. aureus* isolates from Emory University. We genotyped these isolates for the same set of toxins (Table 4). Notably, there were no significant differences between isolates obtained in Iowa compared to Dartmouth or Emory, suggesting that the high prevalence of the EGC is not related to geographic sampling of one region of the United States.

## 4. Discussion

*S. aureus* isolates from patients with CF displayed heterogeneity of color and protease secretion. However, the majority of these diverse *S. aureus* isolates encoded the EGC. The heterogeneity of *S. aureus* colony phenotypes suggests that putative virulence factors such as staphyloxanthin and protease may not be under strong selective pressure to remain in the CF airway. By contrast, alpha-toxin mediated hemolysis was routinely observed. Chronic antibiotic exposure represents a strong selective pressure; most of the isolates from the University of Iowa were resistant to erythromycin. The high prevalence of EGC toxins suggests that *S. aureus* is under pressure to maintain these genes during infection of the CF airway.

We considered the possibility that high prevalence of EGC was related to geographic exposure. Using three geographically distinct collections of *S. aureus* isolated from U.S. patients with cystic fibrosis, we found that EGC toxins had similarly high prevalence. This is similar to a recent study of *S. aureus* isolates from Europe patients with CF, in which EGC toxins were present in ≈57% of isolates [19]. The EGC prevalence within this U.S. collection is even higher, suggesting the continued emergence of strains encoding this locus, possibly through the spread of one or more clones. *S. aureus* clonality is common in CF. In the European study, 5 *spa* types accounted for 25.6% of all *S. aureus* isolates. However, EGC-positive *S. aureus* isolates were not limited to a single clonal group [19]. 

Most of the CF isolates in the current study were obtained in 2011 and 2012, a time period similar to when specimens were collected for a study of atopic dermatitis that used the same genotyping methodology [21]. We compared the prevalence of each of the SAgs in CF versus atopic dermatitis. All of the SAgs belonging to the EGC (*seg*, *sel-i*, *sel-m*, *sel-n*, *sel-o*, and *sel-u*) were significantly more prevalent in CF as compared to atopic dermatitis collection. Among subjects with atopic dermatitis, three distinct genotypes of *S. aureus* were apparent. The first of these *S. aureus* genotypes, which nearly always encoded all EGC toxins, appears highly similar to the prevailing *S. aureus* within the CF population. Longitudinal studies are needed to determine whether or not the EGC associates with persistence of *S. aureus* in diseases like CF and atopic dermatitis.

In comparing the prevalence of the EGC in these CF isolates to *S. aureus* isolates derived from other anatomic sites, we find that CF has a uniquely high prevalence for this group of SAg toxins. The EGC is more prevalent in CF than in atopic dermatitis, diabetic foot ulcer, and significantly more prevalent than in vaginal mucosa [30] and in patients with menstrual toxic shock syndrome (TSS) [12]. The enrichment of EGC-positive isolates in these CF airway isolates suggests that the EGC may confer selective advantage for *S. aureus* strains in adapting to its role as a chronic pathogen in the airway. Typically, members of the EGC are considered colonization SAgs, due to low-level production [31]. 

In striking contrast to the CF, *S. aureus* isolated from acute inflammatory infections such as TSS and post-influenza necrotizing pneumonia have very high prevalence of *tstH* and produce SAgs in higher concentration [12,18]. We observed that a minority of CF clinical isolates encode *tstH*; it is unknown what effect this SAg may have on progression of CF lung disease.

It is unclear whether patients with CF develop intact immune responses to *S. aureus* [32,33]. Compared to *P. aeruginosa*, patients with CF may have attenuated antibody production [33]. We hypothesize that this could be a consequence of immune misdirection by *S. aureus* SAgs. Moreover, these SAgs could facilitate increased inflammation, an important factor in CF disease progression. Future studies should examine the immune response to these prevalent *S. aureus* SAgs. 

*S. aureus* SAgs represent a possible target for vaccination. Given the high prevalence of EGC SAgs in CF, future attempts at immunizing patients with CF against *S. aureus* may use these antigens as vaccine targets. Notably, a recent study has shown that immunization of rabbits against the SAgs TSST-1 and SEC, and the cytotoxin α-toxin protected 87/88 animals after intra-pulmonary challenge [34]. This vaccination strategy depended on formation of cross-protective neutralizing antibodies. For example, antibodies raised against SEC protect against both SEB and the EGC SAg SEl-U. That study also suggested that vaccination against toxoids may be a more effective strategy than vaccination against cell surface *S. aureus* virulence factors. In keeping with this notion, a group has recently performed a first-in-humans vaccine trial against the TSST-1 toxoid [35].

### 4.1. Advantages

This study establishes that enterotoxin gene cluster members are highly prevalent in *S. aureus* isolates in American patients with CF, independent of methicillin resistance. Because we used consistent methodology for genotyping, we can compare respiratory isolates of *S. aureus* from patients with CF to cutaneous isolates from patients with atopic dermatitis that were obtained at a similar time. This comparison reveals that genes encoding EGC toxins are enriched in respiratory isolates. We have also confirmed the high prevalence of EGC toxins using *S. aureus* isolates from geographically distinct CF centers.

### 4.2. Limitations

Because these data are cross-sectional, it is unknown how long these strains have been present in the CF airway. Many of these patients were sputum-producing. Thus, these infections may represent chronic infection rather than initial infection. We are unable to correlate the presence or absence of secreted toxins with pulmonary outcomes such as FEV_1_, since the specimens are de-identified. Although many strains encode secreted toxins, we cannot determine whether these genes are actively expressed in the CF airway. Because of de-identification, we are also unable to determine whether there is adaptive host response to presence or absence of these toxins. We intend to address these limitations in future studies with a larger number of fully identified specimens.

## 5. Conclusions

*S. aureus* remains a prevalent pathogen in CF. Improvements in the prevention and treatment of *S. aureus* infection remain a major goal in this disease. This study reveals the SAg ECG gene cluster is highly prevalent in *S. aureus* CF isolates, revealing a potential vaccination target for this organism in CF. 

## Figures and Tables

**Figure 1 genes-10-01036-f001:**
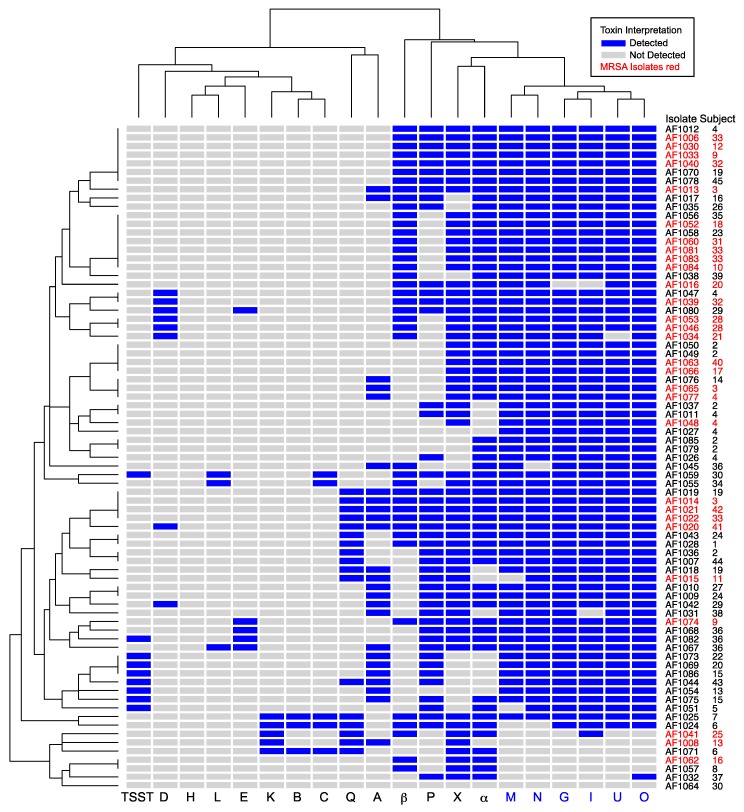
Unsupervised hierarchical clustering of cystic fibrosis (CF) clinical isolates based on presence of *S. aureus* toxin genes. Isolate numbers and subjects are indicated at right. Toxin genes are on the bottom margin, with members of the enterotoxin gene cluster (EGC) in blue. Letters A, B, C, D, E, and G are staphylococcal enterotoxin (SE) superantigens characterized as causing emesis after oral administration. Letters H, I, K, L, M–Q, and X are SE-like superantigens. TSST is toxic shock syndrome toxin-1 superantigen. α and β are cytotoxins. Blue shading represents toxin presence. MRSA isolates are indicated with red font. Dendrograms at left and top show relatedness of the isolates and toxins, respectively. The EGC was prevalent in both MRSA and MSSA isolates.

**Table 1 genes-10-01036-t001:** Phenotypes of *S. aureus* isolated from individuals with cystic fibrosis.

Characteristic	Number of Isolates (Total = 77)	%
**Clinical Source**		
Sputum	44	57.1%
Throat culture	33	42.9%
**Antibiotic Resistance**		
Oxacillin	28	36.4%
Chloramphenicol	0	0.0%
Tetracycline	12	15.6%
Erythromycin	75	97.4%
**Hemolysis**		
Complete, rabbit blood agar ( α-toxin)	63	81.8%
Partial, sheep blood agar (β-toxin)	41	53.2%
**Color**		
White	13	16.9%
Yellow	37	48.1%
Gold	27	35.1%
**Secreted Protease**		
Not detected	41	53.2%
Faint	13	16.9%
Present	23	29.9%

**Table 2 genes-10-01036-t002:** Prevalence of *S. aureus* superantigen genes detected from individuals with cystic fibrosis.

Toxin	Iowa Subjects with CF Total = 45		Single Culture Total = 28		Multiple Cultures Total = 17		*P* ^†^
	N	%	N	%	N	%	
*sea*	19	42.2%	8	28.6%	11	64.7%	0.03
*seb*	2	4.4%	1	3.6%	1	5.9%	1.00
*sec*	4	8.9%	2	7.1%	2	11.8%	0.63
*sed*	6	13.3%	2	7.1%	4	23.5%	0.18
*see*	3	6.7%	0	0.0%	3	17.6%	0.05
***seg***	**42**	**93.3%**	**25**	**89.3%**	**17**	**100.0%**	**0.28**
*sel-h*	0	0.0%	0	0.0%	0	0.0%	-
***sel-i***	**42**	**93.3%**	**25**	**89.3%**	**17**	**100.0%**	**0.28**
*sel-k*	4	8.9%	2	7.1%	2	11.8%	0.63
*sel-l*	3	6.7%	1	3.6%	2	11.8%	0.55
***sel-m***	**39**	**86.7%**	**23**	**82.1%**	**16**	**94.1%**	**0.38**
***sel-n***	**41**	**91.1%**	**25**	**89.3%**	**16**	**94.1%**	**1.00**
***sel-o***	**43**	**95.6%**	**26**	**92.9%**	**17**	**100.0%**	**0.52**
*sel-p*	30	66.7%	15	53.6%	15	88.2%	0.02
*sel-q*	15	33.3%	8	28.6%	7	41.2%	0.52
***sel-u***	**41**	**91.1%**	**24**	**85.7%**	**17**	**100.0%**	**0.28**
*sel-x*	39	86.7%	23	82.1%	16	94.1%	0.38
*tstH*	8	17.8%	3	10.7%	5	29.4%	0.23
≥1 of *egc*	44	97.8%	27	96.4%	17	100.0%	1.00
6 of *egc*	37	82.2%	21	75.0%	16	94.1%	0.13

^†^*P* values compare subjects with single culture to subjects with multiple cultures using Fisher’s exact test. Bold font indicates genes belonging to the enterotoxin gene cluster.

**Table 3 genes-10-01036-t003:** Association of *S. aureus* toxin genes with methicillin susceptibility or resistance.

	*S. aureus* Total = 77		MSSA Total = 49		MRSA Total = 28		OR *	*P* ^†^
Toxin	N	%	N	%	N	%		
*sea*	25	32.5%	17	34.7%	8	28.6%	0.76	0.62
*seb*	3	3.9%	3	6.1%	0	0.0%	-	0.30
*sec*	5	6.5%	5	10.2%	0	0.0%	-	0.15
*sed*	8	10.4%	3	6.1%	5	17.9%	3.28	0.13
*see*	5	6.5%	4	8.2%	1	3.6%	0.42	0.65
***seg***	**69**	**89.6%**	**45**	**91.8%**	**24**	**85.7%**	**0.54**	**0.45**
*sel-h*	0	0.0%	0	0.0%	0	0.0%	-	-
***sel-i***	**69**	**89.6%**	**44**	**89.8%**	**25**	**89.3%**	**0.95**	**1.00**
*sel-k*	5	6.5%	3	6.1%	2	7.1%	1.18	1.00
*sel-l*	3	3.9%	3	6.1%	0	0.0%	-	0.30
***sel-m***	**67**	**87.0%**	**42**	**85.7%**	**25**	**89.3%**	**1.38**	**0.74**
***sel-n***	**68**	**88.3%**	**43**	**87.8%**	**25**	**89.3%**	**1.16**	**1.00**
***sel-o***	**71**	**92.2%**	**46**	**93.9%**	**25**	**89.3%**	**0.55**	**0.66**
*sel-p*	46	59.7%	34	69.4%	12	42.9%	0.34	0.03
*sel-q*	17	22.1%	11	22.4%	6	21.4%	0.94	1.00
***sel-u***	**69**	**89.6%**	**45**	**91.8%**	**24**	**85.7%**	**0.54**	**0.45**
*sel-x*	61	79.2%	33	67.3%	28	100.0%	-	0.0003
*tstH*	9	11.7%	9	18.4%	0	0.0%	-	0.02
*hla*	63	81.8%	37	75.5%	26	92.9%	4.15	0.07
*hlb*	41	53.2%	19	38.8%	22	78.6%	5.65	0.001

* OR = Odds ratio, values > 1 indicate increased odds of the toxin being encoded in MRSA versus methicillin-susceptible *S. aureus* (MSSA). ^†^*P* values calculated by Fisher’s exact test. Bold font indicates genes belonging to the enterotoxin gene cluster.

**Table 4 genes-10-01036-t004:** *S. aureus* toxin genes identified in CF clinical isolates from geographically separate regions. Bold font indicates genes belonging to the enterotoxin gene cluster.

Toxin	Iowa Total = 77		Emory Total = 20		Dartmouth Total = 12	
	N	%	N	%	N	%
*sea*	25	32.5%	0	0.0%	2	16.7%
*seb*	3	3.9%	0	0.0%	0	0.0%
*sec*	5	6.5%	1	5.0%	0	0.0%
*sed*	8	10.4%	3	15.0%	0	0.0%
*see*	5	6.5%	4	20.0%	0	0.0%
***seg***	**69**	**89.6%**	**16**	**80.0%**	**10**	**83.3%**
*sel-h*	0	0.0%	0	0.0%	0	0.0%
***sel-i***	**69**	**89.6%**	**16**	**80.0%**	**10**	**83.3%**
*sel-k*	5	6.5%	1	5.0%	2	16.7%
*sel-l*	3	3.9%	1	5.0%	0	0.0%
***sel-m***	**67**	**87.0%**	**15**	**75.0%**	**9**	**75.0%**
***sel-n***	**68**	**88.3%**	**16**	**80.0%**	**10**	**83.3%**
***sel-o***	**71**	**92.2%**	**15**	**75.0%**	**9**	**75.0%**
*sel-p*	46	59.7%	5	25.0%	0	0.0%
*sel-q*	17	22.1%	2	10.0%	2	16.7%
***sel-u***	**69**	**89.6%**	**16**	**80.0%**	**10**	**83.3%**
*sel-x*	61	79.2%	19	95.0%	11	91.7%
*tstH*	9	11.7%	1	5.0%	2	16.7%

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
