# Peer review of "High Prevalence of Staphylococcus aureus Enterotoxin Gene Cluster Superantigens in Cystic Fibrosis Clinical Isolates"

_genes, 2019, doi:10.3390/genes10121036_

Round 1

Reviewer 1 Report

In the manuscript “High Prevalence of Staphylococcus aureus Enterotoxin Gene Cluster Superantigens in Cystic Fibrosis Clinical Isolates” the authors analyze the prevalence of the isolates encoding bacterial superantigens (Sags), including the enterotoxin gene cluster (EGC), in S. aureus CF isolates from three different clinical centers. In addition, but limited to the isolates from one out of the three centers, phenotypic analysis, such as pigmentation, presence of toxins and antibiotic resistance were performed. The results suggest higher prevalence of the EGC in CF isolates as compare to previously published data on isolates from Atopic Dermatitis (AD) as described in Merriman et al 2016 (ref 21 of the manuscript).

This conclusion is derived from the comparison of original results reported in this paper with already published data on AD isolates. Table 2, contains data from both CF and AD isolates, with the latter already published. Thus, the AD data should be removed from the Results section and discussed in the discussion.

Regarding the comparative analysis between CF and AD isolates, the authors did not consider that in the Merriman et al (2016) paper, the AD isolates were subdivided in three different genotypes (genotype 1-3) and that genotype 1 appears, by analyzing data reported in Merriman (2016), to be more similar to the CF isolates. Thus, the author should compare the prevalence of the genes in the ECG cluster obtained from CF isolates with the different genotypes (1-3) of AD isolates. This could reveal that the enterotoxin gene cluster may confer selective advantage not only in CF but also in AD.

Back to the CF isolates, data to compare Iowa (77 strains) and Dartmouth/Emory isolates are limited to the number of ECG genes detected (Table 4), this appear to be reductive as the complete analysis of the gene panel would better highlight similarities, or differences, among the isolates from different centers. This is a crucial point as in CF the spread of clonal epidemic strains is frequent.

Finally, the authors did not take into consideration the problem of clonality among the isolates which is particularly important in CF and which could directly influence the data reported in this paper (see an example in Garbacz K et al., 2018, Infection and Drug Resistance 2018:11 247–255) This point should be considered.

Minor points

Introduction

Line 29-30. The author statement “Staphylococcus aureus is the most prevalent bacterial pathogen in CF..” should be reformulated taking into consideration the consolidated findings of S. aureus (MSSA and MRSA) and H. influenzae in the early CF lung, whereas P. aeruginosa and Burkholderia cepacian complex dominate in the later stages of disease (Surette MG. 2014. The cystic fibrosis lung microbiome. Ann Am Thorac Soc 11:S61–S65).

Materials and methods

Lane 73-77, the period of bacteria isolation is missing

Lanes 89-91,. Indeed, the authors refer to reference 23 for this methodology but this paper has not open access, making difficult to the reader an easy reading. Additionally, in Merriman et al 2016, which reports data that should have been performed with the same methodology, it is reported that genotyping was performed according to Vu et al 2014. It is mandatory to add all the information that allows the readers to comprehend the results obtained and eventually to use the same methodology to expand the analysis. Therefore we request a brief description of the procedure and the primer used for genotyping.

Some typos errors, such as S. aureus (italics) are present throughout the manuscript; also some references are not reported with the same style.

Results

Table 2, report data relative to the patients, it would be informative to have also the number of isolates per patient, this can be provided as Supplementary materials.

Reviewer 2 Report

Summary: In this study, the authors aim to identify the shared virulence factors in 77 S. aureus clinical isolates collected from 45 patients (both adult and pediatric, median age 15.75 years) with cystic fibrosis that may contribute to the establishment of chronic infection.  They specifically assessed for the presence or absence of 18 superantigens (SAgs).  They compared levels of SAgs in CF airway specimens to skin isolates from a contemporaneous cohort of patients with atopic dermatitis and further compared their findings to CF isolate information collected from geographically distinct centers within the United States. In these isolates, they found a high prevalence of SAg EGC toxins (and sea) which may potentially identify a new target for future vaccination studies and preventative measures. This is an important area of study as identifying strategies to prevent and improve treatment of S. aureus infection is a major goal in the CF community as the authors point out.

Specific Comments/Issues:

As the authors addressed in their discussion, this study is limited as the data is cross-sectional and the samples are de-identified which prevents correlation with important clinical factors such as chronicity of infection, exacerbation history, presence of chronic suppressive antibiotic therapy, FEV1, CFTR genotype, and presence of modulator use. The present study design also prevents further studies exploring host response and the role of adaptive immunity in the establishment of this infection. Again, as the authors point out, this data would be greatly enhanced if they were able to establish if these S. aureus infections are initial or chronic. As noted, subjects provided between 1 and 7 isolates from either expectorated sputum or oropharyngeal swabs and 38 subjects only had a single culture. Clinical information would also be important here as it is rare for a patient to only have one sputum sample collected in a year’s time, just as it is rare for a patient to have 7 samples collected in a one-year span. It would be interesting to see if a sub-analysis on the subjects, albeit a low number, that were able to provide more than 1 sample (n = 17) revealed any patterns or changes in prevalence. Was this done? It would be interesting to look at these virulence facts in comparison to other “controls” in the future – specifically and for example, subjects with chronic nasal or sinus infection with S. aureus or methicillin resistant S. aureus and in isolates obtained from subjects with non-CF bronchiectasis.

The authors do report in their discussion that it is their goal to pursue larger studies with fully identified specimens in the future. Overall, the authors addressed the limitations of their study adequately and identify targets and potential avenues for future studies.
